# Comparison of Compass Suprathreshold Screening Strategies

**DOI:** 10.3390/jcm10194330

**Published:** 2021-09-23

**Authors:** Paolo Fogagnolo, Dario Romano, Giovanni Montesano, Valentino De Ruvo, Luca Mario Rossetti

**Affiliations:** 1Eye Clinic, ASST Santi Paolo e Carlo, Università degli Studi, 20143 Milan, Italy; dario.romano1@unimi.it (D.R.); valentino.deruvo@unimi.it (V.D.R.); luca.rossetti@unimi.it (L.M.R.); 2Optometry and Visual Sciences, University of London, London EC1 0HB, UK; giovmontesano@gmail.com; 3NIHR Biomedical Research Centre, Moorfields Eye Hospital NHS Foundation Trust and UCL Institute of Ophthalmology, London EC1 0HB, UK

**Keywords:** glaucoma, screening, fundus perimetry

## Abstract

Screening programs may be useful to reduce the rate of undetected glaucoma. Compass (CMP, CenterVue, Padova, Italy) Standard Suprathreshold strategy (SST) analyses the visual function at 52 retinal locations. A new Quick Suprathreshold strategy (QST) reduces the number of tested locations to 24. With both tests, the CMP also provides an image of the central retina and a detail of the optic nerve head. The aim of this paper is to measure the performances of SST and QST compared with clinical diagnosis. 63 consecutive healthy subjects and 60 consecutive patients with perimetric defects from glaucoma in both eyes were recruited. They received one test per eye (SST or QST in randomized order); results were classified into normal, suspect and abnormal according to a global index provided in the report and called SupraThreshold Response (STR). Agreement between clinical diagnosis and test output were calculated, and test time was also measured. The agreement with the clinical diagnosis was 95.7% for SST and 95.1% for QST. When two suspect tests were excluded, agreement for QST increased to 96.7%. Test duration was 164 ± 82 s for SST and 71 ± 41 s for QST (*p* < 0.0001). Such a difference was similar in both glaucoma patients (respectively 231 ± 65 s vs. 105 ± 33 s, *p* < 0.0001) and normal subjects (98 ± 17 and 39 ± 9 s, *p* < 0.0001). In conclusion, SST and QST showed similar, high agreement with clinical judgement. Morphological analysis is potentially helpful in further improving the clinical usefulness of suprathreshold tests. QST is characterized by a strong time saving compared with SST.

## 1. Introduction

Glaucoma is the leading cause of irreversible blindness, with more than 70 million people affected in the world and more than 10 million bilaterally blind [1]. As most types of glaucomas are asymptomatic in initial stages, more than half of affected individuals are undetected [2]. Late presentation with the advanced stages of the disease is responsible for late diagnosis and late treatment, with reduced vision-related quality of life, increased social costs [3] and an increased risk of blindness [4].

Even today the identification of glaucoma cases mostly relies on case finding by means of full ophthalmic evaluation and additional tests. Routine tonometry is useful to detect patients with hypertensive glaucoma, but people affected by normal tension glaucoma are frequently missed [5]. Thoughtful examination of the optic nerve head (ONH) is of key importance, but even experts may fail to detect early changes, with an 80% diagnostic accuracy [6]. Additional examinations, such as optical coherence tomography (OCT) of retinal nerve fiber layer or retinal ganglion cells layer and visual field examination are often useful to improve the diagnostic ability [6].

Screening programs have been used in an attempt to improve the diagnostic performances on selected populations and allow early diagnosis and treatment. The question as to which test(s) should be used to obtain the best balance between sensitivity, specificity and cost-effectiveness for glaucoma screening still lacks a definitive answer [7].

The acquisition of optic nerve head photographs and/or retinal nerve fiber layer images are nowadays becoming increasingly popular for screening purposes. Telemedicine has also widened the opportunities in this field: photographs can be easily obtained with non-mydriatic cameras by nurses or assistants and then examined by the physician or even by artificial intelligence [8]. Devices such as optical coherence tomography may also be suitable for screening purposes, as they provide an automatic quantification of structural damage, which, coupled with clinical judgment, can greatly help glaucoma detection [9].

On the other hand, perimetric tests have the advantage of directly exploring visual function, thus providing a better descriptor of the true impact of the disease on patient’s quality of life [10]. Yet functional tests are time consuming and require active participation from screened subjects; examinations often need to be repeated due to the presence of unreliable results in unexperienced subjects (the so called “learning effect”) [11,12]. Fatigue can also affect the quality of the results [13], particularly when long full-threshold strategies are chosen. To reduce test time, fast perimetric strategies have become progressively more widespread. Suprathreshold strategies have been introduced in screening settings. Compass (CMP, CenterVue, Padova, Italy) Standard Suprathreshold (SST) testing is used to perform fast screening for visual field loss, with the advantage of an active retinal tracking, which increases the spatial accuracy of retinal stimulation in case of unstable fixation. The device is also equipped with a confocal imaging system which provides infrared, red-free and true color photographs of the posterior pole, potentially enhancing diagnostic performance [14,15,16,17,18]. To make the screening examination even faster, the Quick Suprathreshold (QST) has been recently introduced: this strategy tests just a subgroup of 24 locations and is expected to take 30 to 90 s per eye. No literature on CMP suprathreshold strategies is currently available. The aim of this paper is to study the performances of SST and QST on a population of glaucoma and normal subjects.

## 2. Materials and Methods

This cross-sectional study was performed at the Eye Clinic of San Paolo Hospital, ASST Santi Paolo e Carlo, University of Milan. The protocol was approved by the institutional review board and carried out in accordance with the tenets of the Declaration of Helsinki. Written informed consent was obtained from each participant.

Consecutive healthy subjects and glaucomatous patients were included between April and June 2021. Glaucoma patients were recruited from the Glaucoma Unit; healthy subjects were chosen among hospital staff and patients’ relatives/spouses. Inclusion criteria were: age of 18 years or more; ability to provide informed consent; best corrected visual acuity (BVCA) > 0.6 in both eyes; refractive error ranging between −10 diopters (D) and +6 D, with maximum 2 D of astigmatism; absence of systemic diseases or use of drugs that could affect the visual field or its correct execution; reliable standard perimetry (CMP 24-2 ZEST): false-positive frequency <18% and the blind spot response frequency <25%.

Additional specific inclusion criteria for the healthy group were: normal ONH in both eyes, as evaluated by means ophthalmoscopy; normal RNFL in both eyes, as evaluated by means of OCT; normal standard perimetry in both eyes (normal mean deviation and pattern standard deviation, and absence of a cluster of three or more contiguous abnormal locations in pattern deviation map); intraocular pressure <21 mmHg in both eyes; no history of ocular trauma, other ocular pathologies and eye surgery except for uncomplicated cataract surgery.

Additional specific inclusion criteria for glaucomatous patients were: presence of glaucomatous changes at ONH or RNFL in both eyes defined by a glaucoma specialist; presence of glaucomatous perimetric defects in both eyes at standard perimetry (abnormal pattern standard deviation or presence of a cluster of three or more contiguous abnormal locations in pattern deviation map); absence of ocular pathologies other than glaucoma; no previous eye surgery except for uncomplicated cataract surgery and glaucoma surgery and no history of trauma.

In order to verify eligibility, patients underwent a complete clinical assessment with BVCA measurement, slit lamp examination of the anterior segment, dilated fundus examination and Goldmann applanation tonometry. Spectral-domain OCT evaluation of the RNFL was performed with Glaucoma Module Premium Edition (GMPE) from Spectralis (Heidelberg Engineering, Heidelberg, Germany) and visual field was tested in both eyes with CMP 24-2 ZEST strategy before inclusion.

Both eyes of the participants were included in the study. Enrolled patients underwent one SST test in one eye and one QST test in the fellow eye in a random sequence (randomization by means of a list of random numbers). No demo program was used as study participants were already experienced with the CMP instrument. In case of unreliable tests (false-positive frequency > 18% and the blind spot response frequency >25%), the test was repeated once; if the subject was able to provide a second reliable test, this was analyzed; in case of a second unreliable result, the eye was excluded from the analysis.

### 2.1. Compass Suprathreshold Strategies

The SST strategy provides a screening and quick assessment of the central retinal sensitivity. The program tests 52 locations regularly spaced by 6° in the central 24° of the visual field with nasal extension to 30°. Each location receives just 2 stimulus projections (“lower intensity” and “higher intensity”, whose intensity is calculated based on values obtained from a normative population). The printout of SST strategy consists of the grid of tested locations superimposed onto a color image of the central retina. Tested points are presented as white circles (if both stimulus have been detected), 50% white-50% black circles (if just the projection at high intensity was detected) or black circles (if none of the two projections was detected). Pointwise responses are merged into a global index called SupraThreshold Response (STR), which represents the percentage of the population of normal subjects having the same result. The index is also shown as color code: green (normality, 11th percentile of normal subjects or better), yellow (suspect, between the 10th and 6th percentile), and red (abnormal, 5th percentile or less).

The new QST tests just a subgroup of 24 locations, mainly located on the central and nasal areas, which are considered the most critical in glaucoma detection. The printout and interpretation of QST are identical to SST. Printout of SST and QST are Given in Figure 1a,b respectively.

### 2.2. Statistical Analysis

The clinical diagnosis (normal or glaucoma) was used as a gold standard. Results of suprathreshold tests were classified according to STR into normal, suspect, and abnormal. The agreement between clinical and CMP strategies was calculated both including and excluding the STR category “suspect”.

Sensitivity and specificity over the study population were also calculated. Again, the analysis was repeated by excluding suspect tests, including them into normal results or including them into abnormal results.

Test duration was also calculated and compared between SST and QST (*t*-test for independent data).

## 3. Results

145 subjects were potentially eligible. 11 glaucoma patients (14 eyes) were excluded due to the presence of other ocular pathologies (ten eyes of five patients), unconfirmed glaucoma diagnosis (absence of visual field damage in glaucoma patients; four eyes of two patients), unreliable standard perimetry (six eyes of three patients; in 4 cases, just one eye was excluded). Four normal subjects were excluded due to the presence of defects at standard perimetry, and four because of unreliable test results (in six cases, just one eye was excluded). Four participants (two normal subjects, two glaucoma patients) were randomized to suprathreshold tests but were excluded because their suprathreshold fields were unreliable. We finally analysed 238 eyes of 123 subjects: 63 healthy subjects and 60 glaucoma patients, whose characteristics are reported in Table 1. Among glaucoma patients, 38 eyes had mild defects (MD higher than −6 dB), 31 moderate defects (MD between −6 and −12 dB), and 48 severe defects (MD lower than −12 dB). 

The results of the study are summarized in Table 2.

The agreement with the clinical diagnosis was 95.7% for SST and 95.1% for QST. When suspect test results were not considered, agreement for QST increases to 96.7%.

SST had sensitivity of 96.5% and specificity of 96.6%. Depending on the interpretation of suspect results in QST, sensitivity ranged from 96.8 to 98.3%, and specificity from 93.6 to 95.1% (Table 3).

Test duration was 164 ± 82 s for SST and 71 ± 41 s for QST (*p* < 0.0001). Such a difference was present also in the subgroups of glaucoma patients (respectively 231 ± 65 s vs. 105 ± 33 s, *p* < 0.0001) and normal subjects (respectively 98 ± 17 s and 39 ± 9 s).

Figure 2 shows the examinations of a patient with glaucoma showing agreement between clinical diagnosis and SST on right eye, and disagreement between clinical diagnosis and QST on left eye.

## 4. Discussion

This is the first study evaluating CMP suprathreshold programs; both screening strategies obtained good performances, as the agreement with the clinical diagnosis was more than 95%. QST also provided a clinically significant advantage in terms of test duration: in both normal subjects and glaucoma patients, QST duration was more than halved.

The main conclusion of this paper is that in the context of a screening program for glaucoma, it may be appropriate to reduce the number of studied locations, as done by QST, to save test time.

This concept contrasts with the current trend in perimetric testing for glaucoma, since many authors have advocated the use of more spatially detailed patterns to enhance diagnostic accuracy and improve detection of change over time [19,20].

In this work we calculated the sensitivity and specificity and this facilitates the comparison with the results of other similar studies on perimetric screening in glaucoma (summarized in Table 4). The performance of CMP suprathreshold tests are apparently superior to other perimetric techniques such as FDT and even OCT.

Yet our results must be considered with caution for several reason. First of all, data on sensitivity and specificity from our work should be used with careful attention on a general population. Using the results of our study (glaucoma prevalence of 50%) on a normal population (prevalence of 2%) would lead to a higher false-positive rate and thus to reduced specificity.

Moreover, we excluded normal subjects showing abnormal sensitivities at standard perimetry and subjects unable to perform reliable perimetric tests; it is probable that this selection further decreased the false-positive rate of CMP suprathreshold procedures. In the context of a “true” screening procedure over a population, this selection would be impossible to perform, and any test defect would lead to positive results—a fact that represent a major limitation for population screening using perimetry. Variability of consecutive results is expected due to the psychophysical nature of perimetry (for FDT, we previously found a disagreement ranging between 2–5% on two consecutive tests) [21]. Finally, our dataset is clearly influenced by the inclusion criteria of the study: we included patients with different severity of the disease (as shown on Table 1), and this may not be representative of the stage of the disease on a general population; moreover, we excluded patients with unilateral glaucoma, who are generally at earlier stages of the disease compared to patients with bilateral involvement.

On the other hand, CMP has the strong advantage of allowing a direct integration between morphology and function by directly inspecting on the same printout a red-free image and a color image of the central retina including the optic nerve head and the perimetric results [18]. Integration of morphology and function increases the diagnostic accuracy of glaucoma diagnosis, both at screening [7] and non-screening levels [17]. This topic could not be formally tested, as there were few cases with incoherent results between clinics and screening tests due to the small cohort of subjects included. Confirmation by large population studies is warranted. Still, the case described in Figure 2 clearly highlights the advantage of adding structural information to the decisional process.

## 5. Conclusions

Our data suggest that CMP suprathreshold strategies show good performance in glaucoma screening, with QST providing similar results with significant time saving. It is likely that even better results may have been achieved if perimetric data were integrated with morphological information.

## Figures and Tables

**Figure 1 jcm-10-04330-f001:**
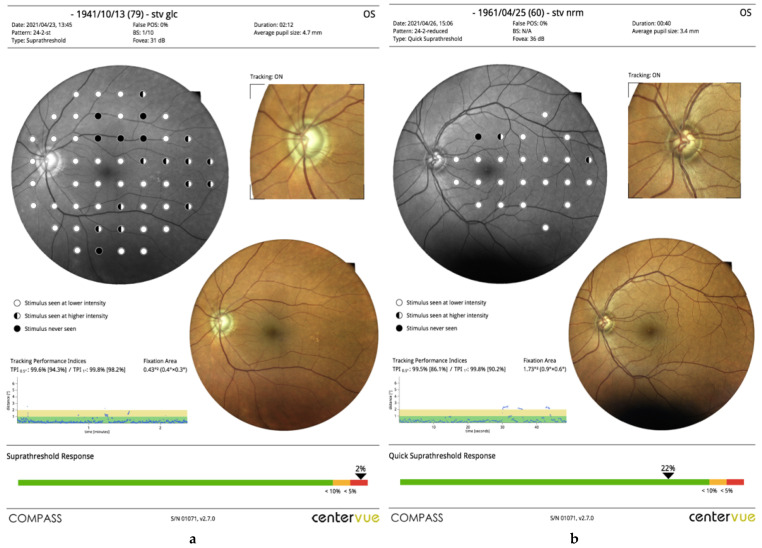
(**a**) This is the printout of SST test. Demographic and reliability data, test time, foveal sensitivity and average pupil size are reported on the top. The main field shows the tested sensitivities superimposed to an infrared image of the central retina; a true-color image of the retina and a detail on optic nerve head are also given. The tracking performance indices are shown. The diagram is useful to inspect the stability of fixation in the course of the test, and the performance of eye movement compensation is given. In the bottom, the suprathreshold response is given using a color scale. (**b**) A printout of QST test is shown. The structure of the printout is identical to SST, apart from a lower number of field locations.

**Figure 2 jcm-10-04330-f002:**
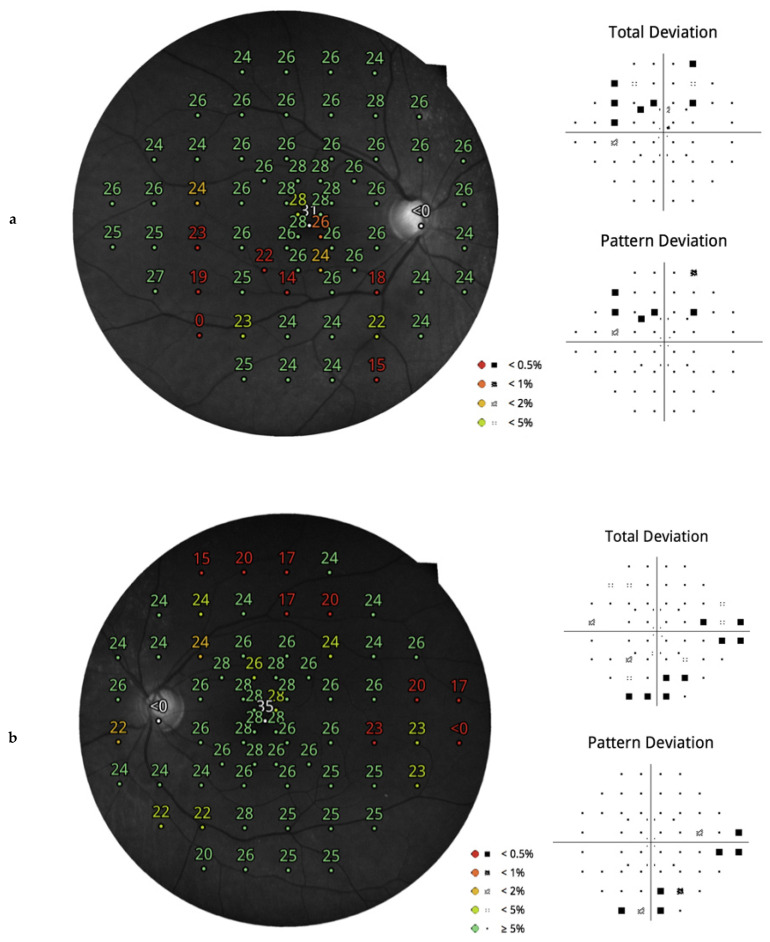
This is the case of a 60-year-old woman with bilateral glaucoma. Standard perimetry showed a bilateral damage (**a**) Right eye, superior relative diffuse defect; (**b**) Left eye, relative superior nasal step and relative inferior arcuate defect). (**c**) OCT showed focal RNFL thinning corresponding to the visual field damage in both eyes. (**d**) The patient received SST in the right eye, and defects at both hemifields were correctly identified; the test was deemed as abnormal (STR = 2%), in agreement with the clinical judgement. (**e**) In the left eye, the patient underwent QST. The screening procedure was able to detect the damage in the superior hemifield shown with standard perimetry; in the inferior hemifield, just an abnormal nasal location was correctly identified, whereas the peripheral abnormal locations of the standard perimetry were undetected. The judgement by STR was borderline (8%). Interestingly, the presence of ONH imaging would strongly allow direct integration of functional and morphological data. (**f**) Inspecting the glaucomatous optic nerve head with RNFL defects (arrows), an ophthalmologist would easily perform a correct assessment of glaucoma.

**Table 1 jcm-10-04330-t001:** Demographics of the study population.

	Normal Subjects	Glaucoma Patients
Number	63	60
Age, years	58.9 ± 17.3	68.4 ± 8.3
Sex, M/F	30/33	33/27
Intraocular pressure, mm Hg	14.0 ± 2.8	14.5 ± 2.6
Mean deviation, MD	+1.75 ± 2.33	−11.94 ± 8.66

**Table 2 jcm-10-04330-t002:** Contingency table between clinical diagnosis and test results.

		QUICK SUPRATHRESHOLD			STANDARD SUPRATHRESHOLD		
		Test Results (*n*)			Test Results (*n*)		
		Normal	Suspect	Abnormal	Normal	Suspect	Abnormal
Clinical diagnosis	Normal	59	1	3	56	0	2
	Glaucoma	1	1	58	2	0	55

The different colors in Table 2 are linked with the color used in the QSR.

**Table 3 jcm-10-04330-t003:** Sensitivity and specificity of QST depending on the interpretation of suspect output of suprathreshold test.

	Interpretation of “Suspect” Results		
	Excluded	Included into “Normal”	Included into “Abnormal”
Sensitivity	98.3%	96.8%	98.3%
Specificity	95.1%	95.1%	93.6%

**Table 4 jcm-10-04330-t004:** Sensitivity and specificity of screening devices for glaucoma detection.

Author, Year	Device	Population	Sensitivity	Specificity
Fogagnolo, 2005 [21]	FDT N-30 screening	*n* = 80 (glaucoma 40, controls 40)	87.5%	95.0%
Kim, 2021 [22]	Red-free fundus photography	*n* = 196 (84 controls, 25 suspects, 87 glaucoma)	92.9%	98.8%
Lee, 2019 [23]	OCT RNFL	*n* = 200 (100 glaucoma, 100 controls)	88%	87%
Lee, 2019 [23]	Deep learning on red-free fundus photography	*n* = 200 (100 glaucoma, 100 controls)	94%	84%

## Data Availability

The data presented in this study are available on request from the corresponding author. The data are not publicly available due to privacy.

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
