# Peer review of "Comparison of Compass Suprathreshold Screening Strategies"

_jcm, 2021, doi:10.3390/jcm10194330_

Round 1

Reviewer 1 Report

The manuscript is clearly written and reads well.

In the discussion authors compare different devices for detection of glaucoma and mention to be cautious about the findings, listing several reasons that are present in clinical practice. Another issue that has not been mentioned is inclusion of glaucoma patients with different severity of disease which affects the results of such studies. I would suggest also mentioning this issue.

Introduction

Page 1; line 33-34: I would suggest rephrasing the sentence. It is late presentation that is responsible for late diagnosis and treatment…

Material and methods

Page 2; line 96: inclusion criteria for glaucomatous patients were presence of VF defect in both eyes. In Abstract it states that patients with glaucomatous VF defect in at least one eye were included.  

Page 3; line 111: “in glaucoma patients only the eye with perimetric damage was used”.

Results

Page 8; line 209: change to “at both hemifields”

Author Response

Response to Reviewer 1

The manuscript is clearly written and reads well.

In the discussion authors compare different devices for detection of glaucoma and mention to be cautious about the findings, listing several reasons that are present in clinical practice. Another issue that has not been mentioned is inclusion of glaucoma patients with different severity of disease which affects the results of such studies. I would suggest also mentioning this issue.

R) We are grateful to the Reviewer for the words of appreciation and the insightful comments. We added on Discussion that: “our dataset is clearly influenced by the inclusion of glaucoma patients with different severity of the disease (as shown on Table 1), which may not be representative of the stage of the disease on a general population”.

Introduction

Page 1; line 33-34: I would suggest rephrasing the sentence. It is late presentation that is responsible for late diagnosis and treatment…

R) Changed as correctly recommended

Material and methods

Page 2; line 96: inclusion criteria for glaucomatous patients were presence of VF defect in both eyes. In Abstract it states that patients with glaucomatous VF defect in at least one eye were included.  

R) I am very sorry for this incongruence and I confirm that all glaucoma patients had bilateral disease. The fact that we excluded unilateral disease may affect the diagnostic ability on a general population and was therefore mentioned in Discussion (“Moreover, we excluded patients with unilateral glaucoma, who are generally at earlier stages of the disease compared to patients with bilateral involvement.”)

Page 3; line 111: “in glaucoma patients only the eye with perimetric damage was used”.

R) This paragraph was modified as follows: “Both eyes of the participants were included in the study. Enrolled patients underwent one SST test in one eye and one QST test in the fellow eye in a random sequence (randomization by means of a list of random numbers). No demo program was used as study participants were already experienced with CMP instrument. In case of unreliable tests (false-positive frequency > 18% and the blind spot response frequency > 25%), the test was repeated once; if the subject was able to provide a second reliable test, this was analyzed; in case of a second unreliable result, the eye was excluded from the analysis.”

Results

Page 8; line 209: change to “at both hemifields”

R) Corrected as recommended

Reviewer 2 Report

This paper describes the analysis of a modestly faster approach for glaucoma screening in a small clinical population. The authors found that decreasing the image acquisition of standard SST still shows relatively good sensitivity and specificity for glaucoma detection. Overall, this is a well-written paper with clearly described background and methods. Suggestions are made to improve the quality of the manuscript. 

-The text in Fig1 is impossible to decipher. Modifications of both Fig1 and 2 are needed to improve readability.

-Please define the differences between Fig1a and 1b in the text. If these are representative images of the study, include this description in the figure legend.

-Please include a standard Table 1 containing all clinical characteristics collected for the purpose of the study, e.g., age, sex, ocular pressure at visit, tonometry measurements, etc.

Author Response

Response to Reviewer 2

This paper describes the analysis of a modestly faster approach for glaucoma screening in a small clinical population. The authors found that decreasing the image acquisition of standard SST still shows relatively good sensitivity and specificity for glaucoma detection. Overall, this is a well-written paper with clearly described background and methods. Suggestions are made to improve the quality of the manuscript. 

-The text in Fig1 is impossible to decipher. Modifications of both Fig1 and 2 are needed to improve readability.

R) I am sorry for this problem which was due to technical reasons (the Figures must be included into the manuscript and not as a separated file). We include the Figures as an attachment; hopefully they will be inserted using proper tools by the editorial office

-Please define the differences between Fig1a and 1b in the text. If these are representative images of the study, include this description in the figure legend.

R) We already reported in the text that SST is shown in Fig 1a and QST in 1b. Clearly, this was not visible due to formatting problems. Sorry for this inconvenience.

-Please include a standard Table 1 containing all clinical characteristics collected for the purpose of the study, e.g., age, sex, ocular pressure at visit, tonometry measurements, etc.

R) Done as recommended.